# The Dutch Flood Protection Programme: Taking Innovations to the Next Level

**Ellen Tromp** [1,2,*] **, Anouk te Nijenhuis** [2] **and Han Knoeff** [1,2]

1. Deltares, 2600 MH Delft, The Netherlands; han.knoeff@deltares.nl
2. Dutch Flood Protection Programme, Location B4, 3500 GE Utrecht, The Netherlands; anouk.te.nijenhuis@rws.nl
* Correspondence: ellen.tromp@deltares.nl

**Abstract:** The Dutch regional water authorities face an enormous task: the strengthening of about 1500 km of dikes and 500 civil-engineering structures before 2050. This immense operation is being funded, prioritised and supported by the Dutch Flood Protection Programme (DFPP), an alliance of regional water authorities and the Ministry of Infrastructure and Water Management. The work will be executed in nearly 300 projects located throughout the country on the coast, lakes and major rivers. To complete this task on time and within budget, innovation (a better insight into the behaviour of flood defences, new techniques and processes) is believed to be the way forward. In this paper, we look at how the DFPP has encouraged innovations between 2012 and the present. We stress the importance of using a sender–receiver approach to further knowledge transfer and uptake, and we describe how, by using an action research approach, the Dutch Flood Protection Programme is currently adapting its innovation strategy on the basis of lessons learned to improve knowledge uptake. We will address some of the innovations that have been developed over the years and how monitoring knowledge uptake helps to further improve the learning-by-doing approach.

**Keywords:** flood risk management; innovations; dikes; flood decision-making; knowledge uptake

## 1. Introduction

Two-thirds of the Netherlands is below the current sea level. In the distant past, the Dutch responded reactively to floods. However, after the catastrophic floods of 1953, the Dutch took measures to prevent a similar disaster: a system of dike rings with flood protection levels was developed for flood risk management [1], and statutory standards were set out in the Flood Protection Act of 1996 to maintain this system. Those standards related to the primary flood defences: those on the coast, major rivers and lakes. In 2008, the second Delta Commission advised changes to take future uncertainties, such as climate change and land subsidence, into account in order to protect the Netherlands.

The Dutch national government recently adopted a risk-based approach [1] for the Dutch flood risk management policy based on new knowledge about the safety of dikes and the impact of serious floods as a result of the failure of dikes and/or other structures. This is a proactive approach, in which protection standards are based on both the probability and the impact of flooding in 2050, considering climate change and socio-economic developments. The Water Act (soon to be the Environment Act) sets out the statutory standards for the flood protection structures, such as dikes, dams, and other civil-engineering structures. Every 12 years, the regional water authorities are required to conduct assessments to ensure that the flood defences still meet the statutory standards. Where the standards are not met, the responsible body can apply for funding from the Dutch Flood Protection Programme (DFPP) [2]. Over 1500 km of dikes and 500 civil-engineering structures will have to be upgraded between now and 2050. The challenge formulated in the overall programme goals for the DFPP is to:

- Increase the production rate (*effectiveness*) of flood defence projects; In programmes in the past, approximately 25 km was upgraded annually. In order to complete the objective of around 1500 km by 2050, this rate will have to rise to an average of ~50 km/year [2].
- Improve *efficiency* by reducing the costs per kilometre; In recent decades, the average cost per kilometre in the Netherlands has increased to more than €10 million (in the 2nd Dutch Flood Protection Programme), and in the early years of the DFPP, the average cost rose towards €18 million per kilometre. Given the annual budget of € 360 million, this figure will have to be reduced again to ~€7 million per kilometre [3].

Although flood risk management is rooted in technical knowledge, the current problems are particularly thorny due to high levels of complexity, uncertainty and conflict [4,5]. The field is complex because flood risk issues are entwined with other local problems involving diverse stakeholders, while the sector is institutionally fragmented, and resources are distributed in a non-hierarchical way [6]. The ever-increasing complexity is producing new challenges and demands.

As regional stakeholders, and society in general, seek to realise additional value (notably, housing and biodiversity) in dikes, the spatial integration of dikes in the local surroundings is becoming more and more important. The Dutch, therefore, changed their approach: from one of coping with water levels associated with specific historical weather events to working with a minimum protection standard based on the probability of flooding and weather events expected in the future, where flood risk management is also an integral part of the living environment [7].

Multiple parties, with their own individual interests and responsibilities, play a role, and they seek to minimise trade-offs. This complexity further emphasises the importance of knowledge management and continuous learning. To achieve its goals, the DFPP recognised that a different approach using new knowledge and innovations was required. The programme has therefore developed a learning-by-doing knowledge and innovation strategy. To understand better why some elements of the knowledge and innovation strategy work and others fail, the authors of the present paper initiated an action-research project to observe, diagnose and intervene in multiple meetings in order to encourage knowledge uptake in dike redesign projects and make recommendations to change the knowledge and innovation strategy.

The paper is structured as follows: Section 2 presents the materials and methods, looking at the Dutch Flood Protection Programme and its knowledge and innovation strategy, followed by a description of the methodology used in our study; in Section 3, we describe how the framework helped to diagnose and remedy problems with knowledge transfer and uptake and the ultimate adoption of the new approach, which is now in use and being monitored. Finally, Section 4 presents the discussion and conclusions of this paper.

## 2. Materials and Methods

### 2.1. Dutch Flood Protection Programme: Context

The present paper looks at the knowledge and innovation strategy of the Dutch Flood Protection Programme (DFPP). In 2012, the DFPP started as an alliance between the national government and regional water authorities. It was the successor to the previous Flood Protection Programme, which was 100% funded by the then Ministry of Transport, Public Works and Water Management. After an evaluation of this programme [8], the funding and organisation became a 50–50 responsibility of the current Ministry of Infrastructure and Water Management and the 21 regional water authorities. As mentioned above, the task facing the DFPP is to upgrade about 1500 km of dikes and about 500 civil-engineering structures before 2050 (see Figure 1). The exact numbers depend on the results of the statutory assessment of flood defences by the regional water authorities. This work involves some 300 projects ranked by urgency on the basis of non-conformity with the statutory standard, and scheduling is based on the available budget of €360 million a year. The DFPP

organisation's primary responsibility is to set up and direct the process of programming and budgeting, to grant subsidies and reporting/accounting. The DFPP also furthers the sharing of knowledge through its own training programme and communities of practice. In addition, the DFPP facilitates the regional water authorities through advisory teams to develop and share knowledge between the DFPP alliance partners. The upgraded flood defences are required to comply with the statutory standards while taking local sustainability and liveability into consideration [2].

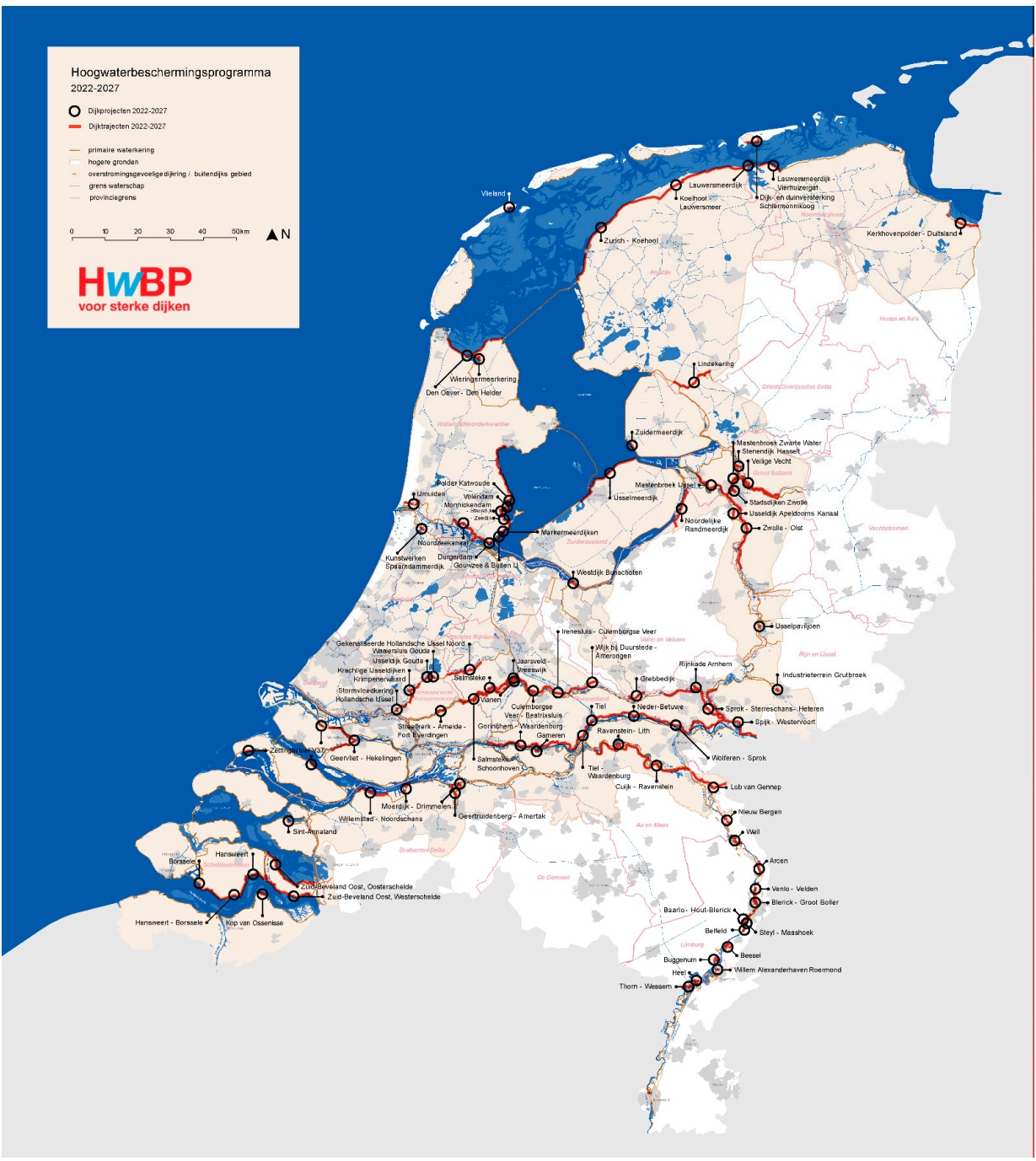

**Figure 1.** Current scope of projects in the Dutch Flood Protection Programme until the year 2027. The expected scope until 2050 is about 1500 km of dikes and about 500 civil-engineering structures.

Innovation is needed to complete the programme on time and within budget, and a knowledge and innovation strategy [3] has been developed for this purpose. The focus is on

two aspects: (1) how to develop a learning-by-doing strategy that takes changes in society over the next few decades into account, since the programme will run for almost 40 years, and (2) how to reduce the development time for innovations further so that knowledge and innovation can be used in the flood defence projects, since the development time in the past has been more than 15 years.

*(1) Adaptive learning by using a Pyramid of Flood Risk Management* [9].

The DFPP developed the Pyramid of Flood Risk Management (Figure 2) to describe the different stages of complexity and the role of flood defence projects in society. The Pyramid of Flood Risk Management identifies four levels, with every level requiring specific knowledge. The next level can only be reached when the preceding level has been adequately mastered. Working towards a new level in the pyramid also requires additional knowledge; the level therefore depends on the context, the actors and the knowledge. The higher the level in the pyramid, the more integrated the approach, which also implies the involvement of an increasing number of actors. Higher up the pyramid, there will be more and more co-creation and actors who reach goals by working together.

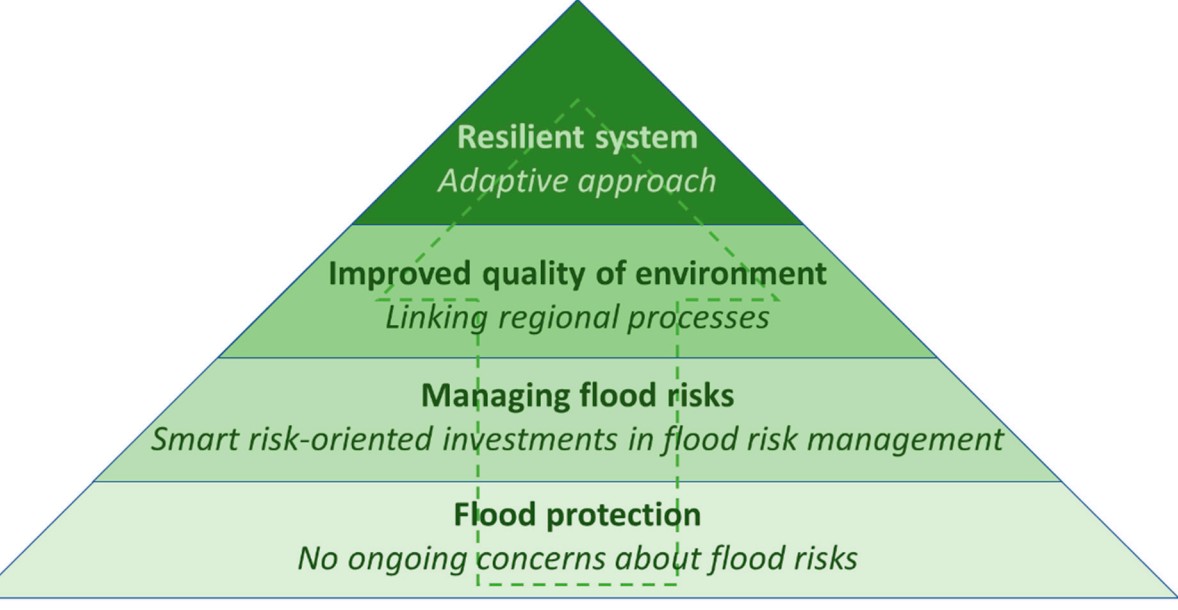

**Figure 2.** The Pyramid of Flood Risk Management.

The lowest level focuses on the behaviour of flood defences, and more specifically, on the different failure mechanisms. Initially, the DFPP focused on developing a better understanding of the different failure mechanisms of flood defences, and on making product innovations for specific failure mechanisms. The recently adopted risk-based approach, as described above, made it possible to move to the second level, 'Managing flood risks'. The evaluations of the successful Room for the River [6] programme contributed skills and knowledge about enhancing the quality of the project area while reducing flood risks. The regional water authorities, especially those active in the Room for the River Programme, adopted the insights acquired in the dike redesign projects under the umbrella of the DFPP. These regional water authorities moved towards the third level in the Pyramid ('Improved quality of environment'). An additional aim of this specific level is to incorporate participatory approaches. Until now, several dike redesign projects have been successful in involving the public [10,11] in order to facilitate the incorporation of ideas and knowledge from local residents and businesses. The top level of the pyramid 'Resilient system' aims to develop adaptive strategies and to reduce future regret relating to investments that must be taken in the coming decades. Linking the implementation of improvements targeting flood risk management to spatial developments and related area processes make synergies and improvements in liveability possible.

Looking through the lens of the pyramid of flood risk management helps the programme to respond to changing boundary conditions associated with society and the climate. This will help to maintain support in Dutch society for flood protection measures. The DFPP aims to develop and further knowledge on all four levels of the pyramid in their innovation portfolio.

*(2) The reduction of development time by using a combination of a stage-gate system and a framework of critical success factors.*

Before the DFPP began, innovators sometimes felt that they had to conduct similar experiments for different organisations to prove their concepts. This is in contrast with the idea that developing successful new products, services or processes is a gradual process of reducing risk by going through several problem-solving phases [12]. Each phase should therefore be designed to reduce the major uncertainties and risks and build upon the previous one. The information required for this, therefore, determines the activities in each phase.

Based on some analysis, the DFPP agreed with the innovators that clarity in the different phases was needed, not least to know how far an innovation was in its development as an accepted alternative. In order to achieve this clarity, the DFPP adopted the Stage-Gate process [13] to make the innovation process as structured as possible, and it developed a guidance document [14,15] to shed light on which knowledge should be available in each phase. This approach helped the DFPP organisation to determine the stage various innovations had reached, but it still did not explain why these innovations did not become mainstream.

To identify why the development of innovations stopped, the DFPP organisation used the critical success factors framework [15] as derived from [16,17]. This framework (Figure 3) identifies six critical success factors:

1.  *People involved*: is sufficient knowledge and expertise available for the project team, and does the involved people have the required competences?
2.  *Instruments*: are the knowledge and tools available to design the flood defences in its surroundings?
3.  *Organisation*: is there political willingness to take risks, and what boundary conditions have been given to the project team? The officials of the regional water authorities are closely involved in the DFPP, through recurring meetings and their role as ambassadors of innovations, which helps in the formulation of the instructions to, and steering of, the project teams.
4.  *Governance*: how is decision-making organised, and how are organisations such as contractors and local stakeholders involved?
5.  *Legislation*: how can the project comply with formal and informal regulations?
6.  *Discourse*: is there sufficient support in society for strengthening the flood defences?

The three success factors on the lower row of Figure 3 are preconditions located outside the boundaries of influence of dike upgrade projects. The remaining three success factors can be controlled by either the regional water authorities or the project team members. This framework helps the DFPP to formulate activities to enhance, and possibly steer, the development of knowledge and innovations.

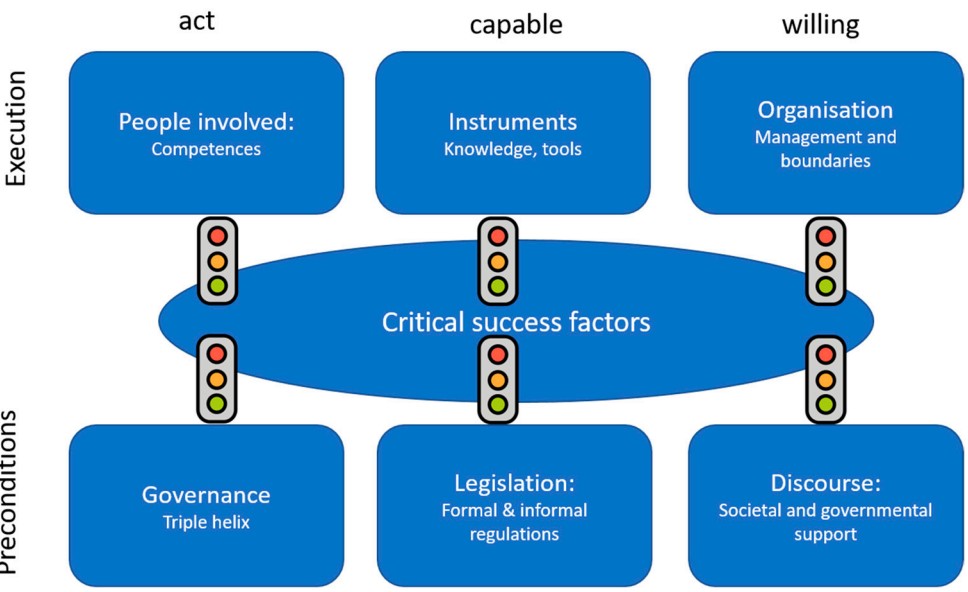

**Figure 3.** The six critical success factors adopted from [15].

In 2013, the aspects of adaptive learning and reduction of development time, as described above, were used to define the scope of seven technical research and test projects. Most DFPP projects focus on redesigning the dikes to address the two most common factors associated with dike failure: slope instability and seepage/piping. Prevention measures are typically expensive and/or require a lot of space. The DFPP, therefore, conducted an assessment of opportunities [2,18] to see which innovations helped to attain the programme objectives. It asked research institutes, regional water authorities and the private sector to propose potential innovations. The innovations were ranked in terms of both the relative performance of projects and the return investment for the DFPP. In particular, the innovations that scored well in terms of the return investment for the DFPP were considered and incorporated into the scope of these seven projects. Some projects, namely Piping, Slope Instability and Wadden Sea, focused on the first level of the Pyramid. Others, such as Forelands, Central Holland and Vechtstromen, focused more on the second level (managing flood risks). The projects not only covered two different levels of the pyramid, they also looked at different critical success factors. For example, Piping focused on instruments and people; Forelands on legislation and organisation and Slope instability more on procurement strategy, governance and instruments.

Several years after the start of the technical research and test projects, the projects were analysed from a sender–receiver perspective using the FODIKI methodology. The analysis, as shown later in this paper, led to greater awareness of the importance of knowledge uptake and continuous learning.

### 2.2. Methods

Our aim was not to evaluate the development of the knowledge and innovation strategy but to ascertain whether the generated knowledge is being applied within the dike redesign projects and whether this leads to different end-user action perspectives and, therefore, helps to relay (i.e., communicate) the objectives. To acquire a better understanding of how knowledge transfer and uptake takes place in the design processes of flood defences, the first author [6] developed and validated a conceptual framework called the Framework for Observing, Diagnosing and Intervening in Knowledge Interaction moments (acronym: FODIKI), as depicted in Figure 4. We assume, as inspired by [19], that knowledge transfer and subsequent uptake entail a knowledge supplier and a knowledge user (from sender S to receiver R). We also retained the possibility of failure (semantic distortion due to cognitive barriers). If a transfer interaction succeeds, Knowledge (K) is available to R, meaning that R can choose to use it. By the 'uptake' of K, we mean knowledge utilisation, as defined by [20]

on a seven-stage cumulative scale, which ranges from reception via cognition, reference, effort, adoption and implementation to impact.

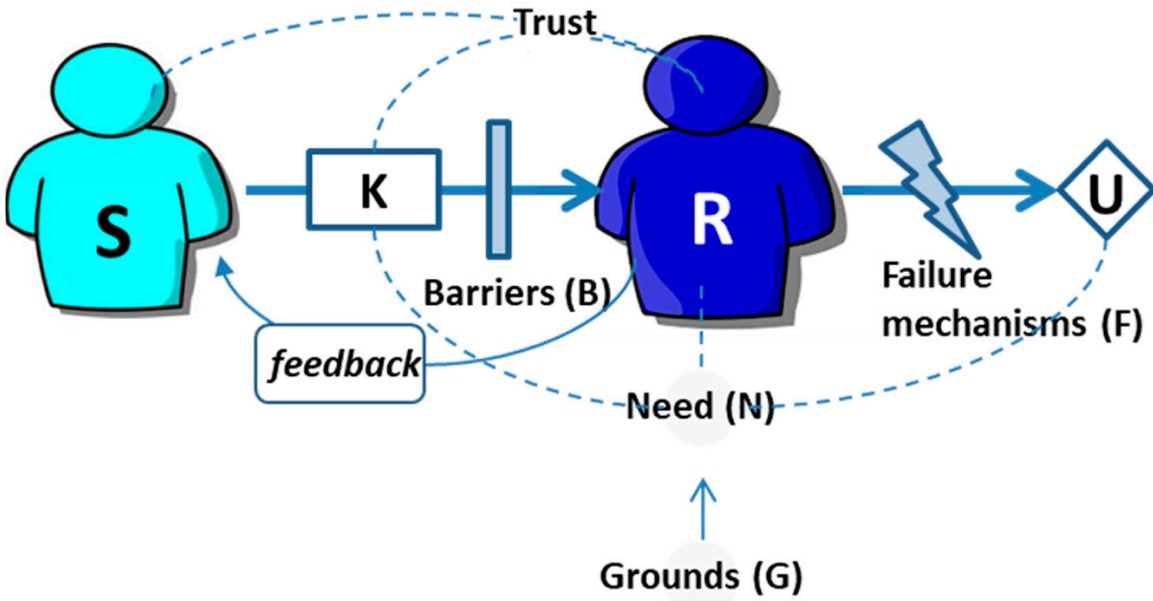

**Figure 4.** A sender–receiver framework for knowledge transfer and uptake [6].

We distinguish three types of social mechanisms that can explain the success of knowledge transfer and uptake: preconditions, barriers and failure mechanisms. Preconditions for communication and reception include a need for knowledge (for which R may have different grounds) and trust (R trusts that S is competent and acting in R's interest; S trusts that R will make good use of K). Barriers hindering knowledge transfer may be transmission (physical barriers hampering communication or poor communication skills), cognitive (R not properly constructing K) and psychological (K conflicting with existing practices or values of R). When barriers of this kind do not arise or can be overcome, knowledge uptake can still fail due to incorrect use (K is used in different ways than intended by S), diffidence (K is disqualified by a third party, dissuading R from uptake) or a lack of relay (R attempts to transfer K to end users but fails). We have also re-introduced the concept of feedback (fb), as both S and R are aware of their roles and the knowledge they have and lack. Through feedback, R is able to communicate on a meta-level whether he understands the shared knowledge and whether a barrier or failure mechanism occurs, after which, S is able to adapt.

We tested the internal validity of the FODIKI methodology by applying it to a historical case study, demonstrating that it allowed for the categorisation and generalisation of observations about the interaction moments of knowledge transfer in a dike design process and an assessment of the actual transfer and uptake of knowledge [7]. To determine external validity, we field tested the FODIKI methodology in "live" cases: (1) a participatory dike redesign process led by a Dutch regional water authority and (2) the knowledge and innovation strategy and related projects in the Dutch Flood Protection Programme. We followed an action research approach for both types of validity.

Action research is characterised by "the active and deliberate self-involvement of the researcher in the context of her investigation" [21]. Researchers and practitioners jointly act in a particular cycle of activities, including problem diagnosis, action intervention and reflective learning [22,23]. Action research typically follows an iterative approach, as depicted in Figure 5, where each iteration cycles through these four steps: plan, act and observe, reflect and then re-plan.

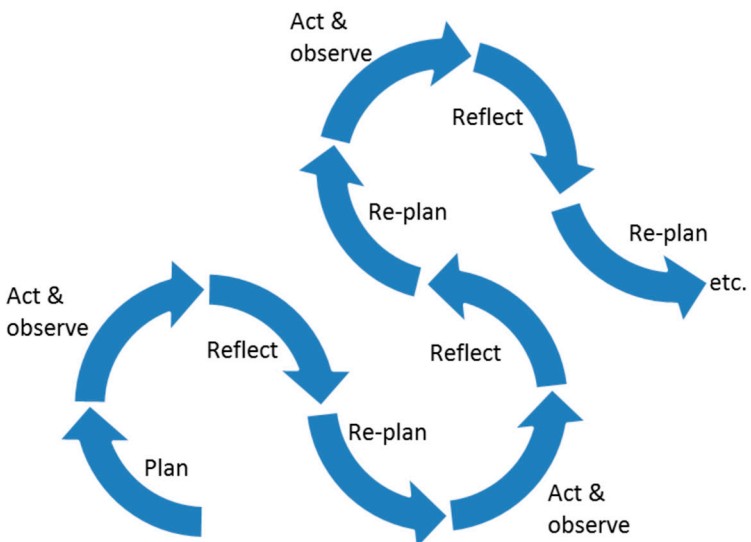

**Figure 5.** Consecutive action research cycles, based on [24].

Here, we focus on knowledge transfer and uptake in the DFPP knowledge and innovation strategy between the developers of the knowledge in the technical research and test projects and the intended end users, in other words, the dike redesign project teams.

*2.3. Data Collection*

We used two different datasets for our study:

(1) Knowledge and Innovation Strategy

Seven technical research and test projects, all led by one of the regional water authorities, began in the period between 2012 and 2018. An initial evaluation was made in 2016 of how four of these projects shared their knowledge, result in reflection and re-planning of the knowledge transfer and uptake. The evaluated projects were Piping, Wadden Sea, Central Holland and Slope stability. The state of knowledge creation, transfer and uptake in four projects was analysed by means of a document analysis and in-depth, semi-structured interviews with the professionals involved. For each project, we interviewed four professionals and shared our findings in a concluding workshop to share the insights and jointly draw up recommendations [25]. This was the first action research cycle.

These findings led to a second action research cycle, with a specific focus on the amendments for the knowledge and innovation strategy. The new knowledge and innovation strategy was shaped in a series of 10 meetings with stakeholders from the Ministry, regional water authorities, research institutes and representatives from the contractors and engineering companies. These meetings were well documented and actively shared within the alliance to ensure that feedback and input was given throughout the process. Not only did representatives from the administrative level attend, the officials of the Ministry and the regional water authorities were also informed and challenged to be part of this new strategy in a series of five meetings [26].

The third action research cycle comprised six meetings to develop the current knowledge and innovation strategy. These meetings were documented [6,27,28], and using the FODIKI methodology, the observations were diagnosed and suggestions for interventions were given. Representatives from the regional water authorities, contractors, engineering companies and research institutes were consulted about the different elements of the new knowledge and innovation strategy. Project team members of the seven technical research and test projects reflected on the new elements to pinpoint where further improvement was feasible.

(2) Monitoring data

Beginning in 2018, the DFPP organised annual knowledge and innovation monitoring, in which they reported how much is spent on knowledge and innovation and what the

outputs, outcomes and impact are. Monitoring took the form of a survey. In the first years, only the knowledge and innovation projects fill in a questionnaire about their progress, how they share their knowledge and what the outputs and outcomes are. Questions were asked how the created knowledge was shared and if reinforcement projects also used that knowledge. The potential savings of the reinforcement projects were requested, both at the project- and programme-scale. In 2021, the survey was sent to stakeholders from research institutes, regional water authorities and the Ministry but also to contractors and engineering companies. One objective of the survey was to find out whether certain instruments were known. The survey included questions such as:

- Which knowledge and innovations do you know or have you heard of?
- Which innovations are being considered in your project?
- Which innovations are being developed within your project?
- Which new knowledge and innovations are/will be applied in your project?
- What are the main reasons for considering or applying all the knowledge and innovations you mentioned in your project?

In the multiple-choice answers, the six critical success factors could be recognized, without being explicitly mentioned. Furthermore, the respondents were asked whether the application of new knowledge/innovations would lead to savings in time and budget for their project. Finally, the respondents were asked which new knowledge and/or innovations could make the greatest contribution towards achieving the overall programme goals.

The results were used for the action research cycle to define subsequent steps. The results from 2018 [29] and 2019 [30] were used in the evaluation of the strategy in 2020, and so there was no monitoring in 2020. The most recent survey from 2021 [31] was used to pinpoint how to optimise the instruments available at the DFPP to share and absorb knowledge.

## 3. Results

### 3.1. Knowledge and Innovation Strategy

In this section, we will describe, using the FODIKI methodology, the three action research cycle loops that we have observed, diagnosed and adapted.

1.   Action research cycle: Analysis of four technical research and test projects

Three years after the start of the projects, we conducted our in-depth analysis [6,25], first, by analysing the documents and, then, by conducting in-depth semi-structured interviews with team members and intended end users. Our observations showed that the teams produced different types of knowledge, such as methodological and process knowledge. The knowledge was, in some cases, also developed for, and used in, dike redesign projects. Knowledge uptake in those cases was high. After our first observations and diagnosis, we shared the preliminary findings with the stakeholders involved to contribute to the learning-by-doing approach.

Our diagnosis revealed that the project teams were creating new knowledge but did not see themselves as senders of that knowledge. Some professionals believed that 'the knowledge will sell itself'. However, since the end users had not been identified, it was difficult to let the knowledge flow. The project teams did not involve the end users, which meant that the intended receivers did not automatically trust the knowledge or that the developed knowledge met their needs.

The diagnosis also highlighted the importance of *boundary spanners*, both for the projects and the overall programme. Boundary spanners are [32] "people who proactively scan the organisational environment, employ activities to cross organisational or institutional boundaries, generate and mediate the information flow, coordinate between their 'home' organisation or organisational unit and its environment and connect processes and actors across these boundaries". The boundary spanners play an active role in sense-making and identifying the knowledge needs of the current and future receivers in the DFPP.

Furthermore, we found that in projects where co-creation took place with the dike redesign projects, the knowledge was appropriate to the knowledge needs and was easily absorbed in the project, leading to higher levels of knowledge uptake. In addition, these projects actively shared their insights with interested colleagues, playing their boundary spanning role.

We proposed several interventions to enhance knowledge transfer and uptake, with the most striking being the development of *knowledge strategies* for each project to identify the intended receivers and their potential needs and their knowledge networks. Another intervention was that the project team members also fulfilled an ambassador's role with respect to their own mother organisations in order to share the developed knowledge.

2.   Action research cycle: Innovation Next Level

Despite all the efforts made, knowledge uptake from these technical research and test projects remained relatively low. The DFPP organisation undertook action and initiated a new action research cycle, in which the researchers helped to shape the successive steps. In this cycle, 15 meetings were organised, of which five were exclusively for the officials of the Ministry and the regional water authorities. Each meeting was thoroughly prepared using the FODIKI methodology, and each successive meeting used the diagnosis and proposed interventions of the previous meetings [27]. In the first 10 meetings, representatives from regional water authorities, the Ministry, engineering companies, contractors and research institutes were invited to share their experiences and suggestions for improvement. The DFPP organisation was able to identify the critical success factors and needs and to build trust to move the knowledge and innovation strategy towards the 'Innovation Next Level'. The DFPP organisation identified a number of shortcomings resulting in a failure to absorb knowledge. They included the absence of a 'risk safety net' in response to the fact that regional water authorities said that they were hesitant to use innovations when the risk of the innovation not working was not covered by the subsidy arrangements. The DFPP had foreseen this risk and already included this risk in the subsidy, meaning that, if an innovation failed to work, the regional water authority could call on the DFPP to cost the reinforcement of the defence.

In a series of five meetings with the officials of the Ministry and the regional water authorities, the officials came to the understanding that 'the baby should not be thrown out with the bathwater'. They agreed that more efforts had to be made to share the acquired knowledge with current and future projects in order to valorise its potential. This is linked to the 'no relay' failure mechanism in the FODIKI methodology. The intervention proposed was to draw more and more on the different DFPP communities of practices, such as the Technical Manager community and Project managers community, in order to share the developed knowledge and to update the training programme. The officials also foresaw a role for themselves as ambassadors for the developed knowledge.

3.   Action research cycle: Towards a new Knowledge and Innovation Strategy

In a third iterative action research cycle, the DFPP organised six consecutive meetings to develop a new knowledge and innovation strategy that focused on transparency, effectiveness and a stronger coupling of the research project and dike redesign projects [27,28]. It built on the insights acquired during the second action research cycle. After each meeting the team reflected and made an analysis based on their observations and diagnosis of how the strategy could be adapted to the needs of the end users. This led to interventions in the strategy (i.e., adaptations), which were discussed with the same end users but also with a new range of end users, slowly converging to a widely accepted approach and resulting in adaptive ongoing development, as depicted in Figure 6. A more detailed description of the new strategy can be found below the figure.

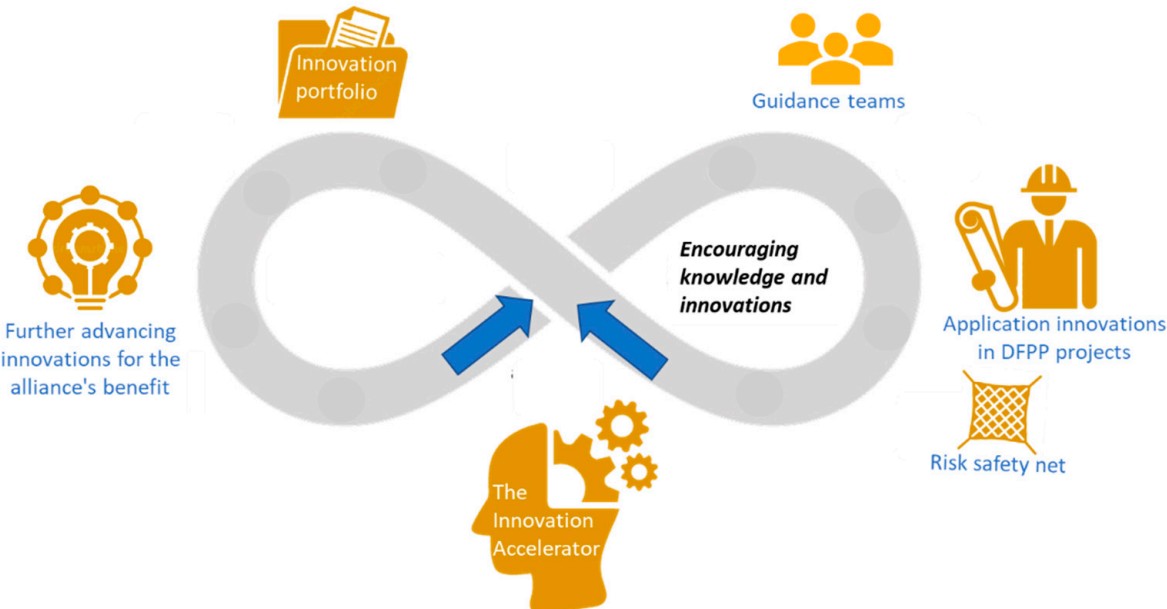

**Figure 6.** Adaptive ongoing development of knowledge and innovations and encouragement of the use of knowledge and innovation in the Dutch Flood Protection Programme.

In the new knowledge and innovation strategy [9], the DFPP highlights three relevant themes that are linked to the Pyramid of Flood Risk Management. The first theme (*technology)* is linked to the first and second layer of the Pyramid of Flood Risk Management, and it focuses on furthering understanding of how flood defences fail and linking that understanding to the new flood protection standards based on the probability of flooding. These innovations aim to limit the scope of the overall programme and to generate new dike reinforcement techniques [9]. The *integrality* theme (which is linked to the third layer of the Pyramid) tackles broader topics relating to, for example, sustainability, integration and the project approach. The last theme looks further ahead and targets the preparation of the programme for future developments: dealing with increased climate change and new societal challenges. These three themes should help the DFPP to remain adaptive and flexible in a context of change.

The DFPP organisation developed a programmatic approach, orchestrating activities so that the programme objectives would be met. The benefit of the knowledge and innovation developments is often not to be found at the individual project level, but rather at the overall programme level, since successive projects are needed to develop innovations and to reap the benefits in terms of costs, time and quality. To support this programmatic approach, the DFPP initiated (1) a structural alliance with several parties, including research institutes, engineering companies and contractors; (2) the 'risk safety net', referred to above, to ensure that risks are 100% covered by the alliance and (3) an Ambassadors Group to establish support from society at large.

To safeguard an ongoing knowledge and innovation process, several instruments were developed (1) to further the continuous development of knowledge and innovations to meet the overall programme objectives, (2) to encourage the application of new techniques and measures and (3) to share developed knowledge and experiences.

1.    Furthering innovations

An R&D budget of €10 million a year is available (2.5% of the total annual budget of DFPP) to encourage the development of innovations. Innovation projects are 100% subsidised. In addition, the risk safety net referred to above ensures that 100% of the risks of innovations are borne by the alliance. Anyone from the regional water authorities, contractors, engineering companies or research institutes can submit ideas to the DFPP organisation to benefit from the R&D budget. For an idea to be included in the innovation

portfolio, a regional water authority has to be involved, as only they can apply for funding. The DFPP organisation orchestrates the involvement of one or more regional water authorities on the development of ideas with potential at the programme level. When assessing the potential of ideas, the DFPP organisation focuses on the following criteria:

- Return on investment at the project and programme levels (balance of the costs and benefits between the project and programme);
- Reproduction factor (level of development in relation to applications in the programme);
- Contribution to alliance (builds on previous projects and seeks cooperation with other water authorities and other knowledge developers).

The involvement of the regional water authorities also safeguards ownership and practical application, as innovation is not a goal in itself, but a way of meeting the overall programme objectives. After the approval of an idea for the innovation portfolio, the innovation projects are carried out, often in conjunction with dike redesign projects, as this stimulates the knowledge uptake.

2. Encouraging the application of new knowledge and innovations

The DFPP has developed a *comply or explain* instrument, which consists of an overview of the new knowledge available and the development level of innovations, as well as a framework for considering application in projects. The overview contains a list of knowledge and innovation developments with a potential impact on the programme. When the potential impact has been validated and the applicability of the innovation has been described, projects are asked to contribute to the further development of the innovation. When the application of innovations and knowledge has been demonstrated in practice, projects are asked to take these into account as accepted alternatives. The overview of available knowledge and innovations is managed by the Innovation Accelerator, which consists of professionals from regional water authorities and research institutes. In addition to maintaining an overall perspective, the Innovation Accelerator helps dike redesign projects with the application of accepted innovations, and it translates project results into, and implements, generically applicable tools.

In each phase of the project, the dike redesign project teams and the advisory team examine which knowledge and innovations can be applied or further developed in the reinforcement project. First, the project team selects promising new knowledge and innovations and determines the contribution of the new knowledge and innovations to the project and programme objectives. This is a cyclical process, which means that it is repeated again and again in every project phase. The project team also makes agreements with the advisory team about how risks can be addressed in the reinforcement project but also about how the developed knowledge can be shared widely within the alliance to encourage large-scale application.

After the completion of a dike redesign project, risks may still arise. For example, the costs in the management and maintenance phase may be significantly higher than expected, or the implemented technique may not achieve the intended safety level, and so the primary flood defence will have to be upgraded again with a different technique. The risk safety net referred to above can be used in cases like these.

3. Knowledge transfer and uptake

For the innovations to be effective, the knowledge obtained in the innovation projects must flow to ongoing and upcoming dike redesign projects. Successful knowledge transfer and uptake means that the knowledge becomes common property within the alliance. Different resources are at the disposal of the DFPP to enhance this process. For example, the DFPP organisation increasingly uses boundary spanners to help them and the projects. The lessons learned from the initial period [6,25] helped to further define the role of the boundary spanners, who can act as brokers, translators or synthesisers:

1. The broker role matches the supply and demand of knowledge. This requires a state-of-the-art overview of existing knowledge and the current knowledge gaps.
2. The translator role interprets the knowledge needs of the end users and formulates questions for the knowledge developers. The translator is also capable of translating the knowledge to match the needs of the end users.
3. The synthesiser is capable of synchronising knowledge supply and demand. This requires a broad understanding of knowledge disciplines and sources and of how they can contribute to problem-solving.

Examples of boundary spanners fulfilling all three roles between innovation and strengthening projects are both the members of the Innovation Accelerator and the advisory teams of the DFPP organisation. To further enhance the knowledge transfer and uptake of the gained knowledge in the innovation portfolio, the DFPP organisation started to monitor the innovation projects in the Innovation portfolio in 2018.

### 3.2. Monitoring Knowledge and Innovation

Before turning to the monitoring of knowledge and innovations, we will start by giving an overview of the knowledge and innovation projects that took place since 2012 in the DFPP. Please note that the focus slowly shifted over the years to other aspects as society also changed over time. The focus of the first DFPP knowledge and innovation projects (Piping, Central Holland) was on gaining a better understanding of the implications of the flood protection standards [2]. A clearer understanding of the behaviour of flood defences was necessary for an effective approach to reinforcement. The slope instability project developed a step-by-step plan for the detailed modelling of the strength of a flood defence. This involved using specific soil investigation and monitoring and advanced calculation techniques, including probabilistic failure probability analyses and failure probability updating, to estimate the actual current strength for the slope-instability failure mechanism as closely as possible [33–37]. Applying the step-by-step plan reduced the need for reinforcement on many projects, and at some locations, reinforcement proved to no longer be necessary. Various studies have also been carried out on piping [38,39] and the deterioration of revetments [40,41] (see also Figure 7b), making safety factors explicit, so that the need for reinforcement can be determined more accurately. Furthering the knowledge uptake will limit the scope of the programme

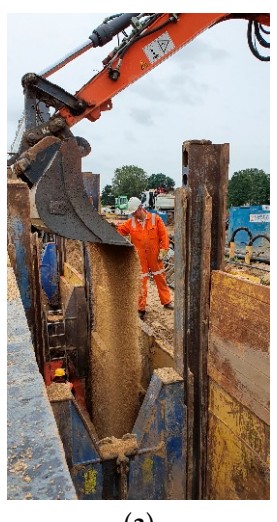 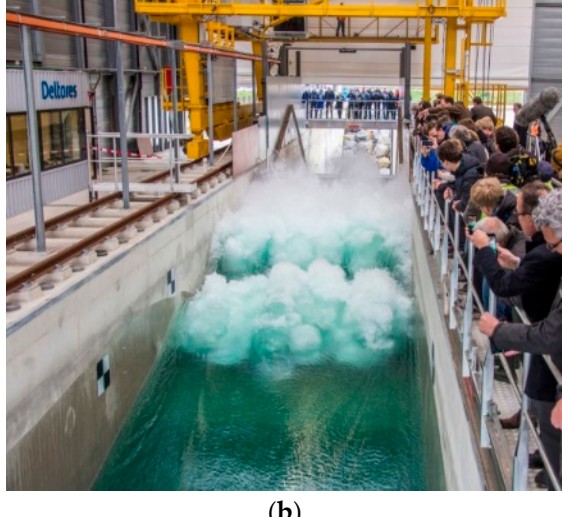

(**a**)                                                                 (**b**)

**Figure 7.** Examples of innovations in the Dutch Flood Protection Programme. (**a**) Implementation of the Coarse Sand Barrier in the Waalban dike in Gameren (photo credit E. Tromp); (**b**) testing in the Delta Flume at Deltares for new type of dike revetment in the northern part of the Netherlands (photo credit: Deltares).

In addition, enhancing knowledge uptake paves the way to optimise product innovations. Traditionally, for instance, resistance to piping is reduced by constructing verges or, when there is not enough space, with sheet piling. Various alternative measures that are more economical and sustainable have been developed in the DFPP, e.g., the vertically inserted geotextile. These experiences led to a modified technique, namely the coarse sand barrier [42,43]. This involves introducing a vertical trench with coarse sand at the transition between the clay layer and the sand substrate that replaces the original finer sand. The coarse sand acts as a filter that lets the water through and keeps the sand grains in place, preventing piping. After a successful series of tests, the Rivierenland Regional Water Authority applied a coarse sand barrier for the first time in the summer of 2021 in the Waalban dike in Gameren (Figure 7a). A better understanding of the behaviour of the defences will thereby lead to a further downscaling of the measures to be taken and, therefore, to the optimisation of the budget, the planning and the impact.

From 2016 onwards, the innovation portfolio increasingly included innovations that help to improve the quality of the living environment (slowly moving up the Pyramid of Flood Risk Management). In the Vechtstromen project [44], a system approach was developed that allows measures that retain water upstream to be combined with dike reinforcements downstream. The Forelands project [45] studied how forelands could be used to reduce flood risks. In this project, they not only examined 'physical' measures, but also looked at the opportunities for, and impediments to, legislation and regulations. These two projects led the DFPP organisation to realise that non-technical themes are equally important, resulting in new types of projects: projects with (1) an emphasis on learning and cooperation, such as organising asset management, and (2) reinforcing dikes in areas covered by legislation on nature and biodiversity. In 2020, a roadmap was drawn up to examine a range of sustainability issues in conjunction with each other in the years ahead [46]. The roadmap also opened up the road to the further investigation and use of the potential of nature-based solutions [47–49], as society calls for.

To determine whether the current innovation portfolio will allow the DFPP to meet the overall programme objectives (more economical and faster upgrades), the DFPP organisation engages in monitoring activities. The aim is to evaluate the knowledge and innovation approach and the annual expenditure on, and commitment to, knowledge and innovation in the DFPP programme. In addition, the DFPP organisation wants to determine what is going well, the trends in place and where improvement is possible. Monitoring took place in 2018, 2019 and 2021.

Monitoring has taken the form of a survey since 2018. In the initial years, innovation projects and dike redesign projects with innovations were asked to complete the survey. In 2021, the survey was sent to stakeholders from governmental organisations, research institutes and contractors and engineering companies. This change in the target group led to a higher response rate and a doubling of the number of respondents from dike redesign projects [31].

In 2018, the respondents stated that working across organisations (56%) and putting innovation at the centre (48%) helps to achieve the goals of the DFPP [29]. In 2019, it was concluded that the development of innovations was accelerating [30]. The first successes work like a flywheel: projects and organisations feel more and more responsible for using and further developing the new knowledge from the innovation projects. The mindset is changing: the ever-smarter implementation of dike redesign projects has now become part of the regular activities. The assessment of opportunities is currently a standard component of the dike redesign process by the following regional water authorities: Rivierenland, Aa en Maas, Vallei en Veluwe, Zuiderzeeland and Stichtse Rijnlanden. Moreover, knowledge is being actively disclosed both in- and outside the alliance through reports, articles, contributions to training courses and presentations in the DFPP communities of practice, consisting of representatives from the regional water authorities, engineering companies and research institutes.

On the basis of the monitoring in 2021, it was concluded that innovations are becoming more widely known and being applied more often: approximately 95% of the respondents were familiar with one or more innovations from the Innovation portfolio. On average, current redesign projects are considering nine innovations, developing three and applying four. In 2018 and 2019, innovations were considered and applied less often, and the emphasis was on contributing to the development of knowledge and innovations. In 2021, fewer barriers were identified: if innovations are not applied, it is often because traditional solutions suffice or because time and capacity are lacking.

It was concluded from the 2021 monitoring that innovation continues to have a positive image and is yielding benefits. Over 50% of the respondents in 2021 reported that the knowledge and innovation projects led to a reduction of the amount of upgrade work needed, better integration, more support and savings and an impact on emissions and circularity. Still, innovation takes time: many innovations are known and considered but not yet applied by everyone. This shows that there is potential for the future, but it also demonstrates the importance of the *comply or explain* instrument. However, half of the respondents are unaware that *comply or explain* and the Innovation Accelerator contribute to innovation. Furthering these instruments is required, especially as other instruments, such as 100% subsidies, the use of advisory teams and the risk safety net, are already seen by some 70% as supportive.

The investment in knowledge development and innovations is yielding results. The total investment in the innovation portfolio was €139 million for 2014 to 2021, inclusive. The direct savings reported for 2020 and 2021 amounted to over €200 million. These are in addition to the savings reported in 2019 for the period of 2014 to 2019 of €170 million, and they bring the total of quantitative savings reported to approximately €370 million. However, not all projects report the benefits of innovations for reasons such as not being able to unambiguously attribute the savings to innovation, uncertainty about whether the savings will be realised or the difficulty of quantifying the savings (which are often avoidable costs). Given the actual and planned expenditure of €4.3 billion until 2030, a saving of 20% is expected from the currently known innovations [31]. Continuous emphasis on knowledge transfer and uptake is essential to monetise it.

## 4. Discussion and Conclusions

In the decades to come, the Dutch regional water authorities and the Ministry face a challenging task to reinforce about 1500 km of dikes and 500 civil-engineering structures. Especially given the recent insights [50] that climate change is accelerating at a higher rate than thought, low-lying populated areas are exposed to an even higher risk than initially believed [51]. The Netherlands, like many other countries, is facing uncertain challenges relating to housing, biodiversity and the energy and agricultural transitions. These challenges, which include climate change, will undoubtedly have an impact on the implementation of the Dutch Flood Protection programme (DFPP). Society's norms and values will also change over the next three decades. The learning-by-doing knowledge and innovation strategy adopted by the DFPP will allow it to be adaptive and flexible. As knowledge is situated and socially constructed, this knowledge must be actively shared and restated after each change in the group of participants. At the same time, a constant awareness of the impact of change on the programme's implementation will be required, as well as additional knowledge. This could also affect how knowledge is shared and absorbed in the future.

Literature shows that various forms of participatory water management [52] have been developed to better handle the wide variety of problem perceptions, values and knowledge. Their effectiveness has been researched from different perspectives, notably social interaction [53,54] and uncertainty [55]. In this paper, we used a knowledge management perspective. Our observations showed that the FODIKI framework helps in detecting and diagnosing social mechanisms in a variety of settings, thus encouraging knowledge uptake within the DFPP, specifically in the knowledge and innovation strategy.

If we interpret our observations on knowledge development, transfer and uptake in terms of team learning and organisational learning [56], we see a lot of substantive team learning and boundary spanning. In recent years, the role of the DFPP as a boundary organisation became apparent, given its links with the regional water authorities and their projects and with policy and practice. Working practices at the DFPP have changed, and knowledge management is being developed further. The role of boundary spanners is being taken more seriously to ensure that the institutional knowledge and information about precedents in dike redesign projects are kept within the DFPP organisation in order to achieve the programme goals. The role of the advisory teams as boundary spanners is still in its infancy: they are still learning on the job, and knowledge sharing between advisory teams is limited. Ongoing attention is needed to further encourage this sharing and help the teams grow into their role. At the same time, the DFPP organisation is encouraging interorganisational learning between the projects and regional water authorities involved. The question arises of how knowledge uptake can be secured for the long term. Staff turnover in a knowledge-intensive process often has a demoralising effect on others. Given the fact that most regional water authorities will work on projects for the next three decades, the question arises of how is the current and future available knowledge being secured for this task, both at the regional water authorities and at the DFPP organisation? This is because like the DFPP organisation, a specific department at a regional water authority carries out the dike redesign projects and also acts as a boundary organisation. This means that regional water authorities need to devote attention to encouraging both organisational and interorganisational learning. The different organisations therefore need to maintain a focus on the role of boundary spanners. Efforts must be made to ensure that the parties involved have the knowledge-uptake capacity that is required. However, earlier research [7] has shown that sharing knowledge between several projects at one regional water authority is already difficult.

In this paper, we have described how the Dutch Flood Protection Programme has adopted a Pyramid of Flood Risk Management, a stage-gate system and a framework for critical success factors. Our study has shown that, given the current time frame, the pyramid of flood risk management and the critical success factors framework are helpful in terms of the observations, diagnosis and interventions for the strategy. However, these concepts have been loosely adopted with no methodological rigour. Further research on the validity and reliability of these concepts is required to use them for the long term.

To conclude, the new knowledge and innovation strategy appears to be working given the increased levels of knowledge uptake. Nevertheless, the DFPP organisation has been monitoring knowledge uptake for only three years, and it has made interventions based on the results. This period is relatively short as a basis for concluding that the current knowledge and innovation strategy is working and that knowledge uptake is sufficient to meet the programme goals, particularly because the pilot paradox [57,58] is still hanging over the DFPP: previous research showed that a successful pilot project (which delivers useful knowledge and results in more knowledgeable participants) is anything but a recipe for successful knowledge uptake and upscaling. To bridge this gap, the DFPP needs to involve future users in early phases of the project, and it should start communicating some of the successes and obstacles to a wider public early in the innovation projects.

**Author Contributions:** Conceptualization, E.T. and H.K.; methodology, E.T. and H.K.; validation, E.T., H.K. and A.t.N.; resources, E.T., H.K. and A.t.N.; data curation, E.T., H.K. and A.t.N.; writing— original draft preparation, E.T. and H.K.; writing—review and editing, H.K. and A.t.N.; visualization, E.T.; supervision, A.t.N. All authors have read and agreed to the published version of the manuscript.

**Funding:** This research received no external funding.

**Institutional Review Board Statement:** Not applicable.

**Informed Consent Statement:** Not applicable.

**Data Availability Statement:** Not applicable.

**Acknowledgments:** The authors would like to thank the reviewers and editors for their critical and targeted comments.

**Conflicts of Interest:** The authors declare no conflict of interest.

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
