# Peer review of "The Dutch Flood Protection Programme: Taking Innovations to the Next Level"

_water, doi:10.3390/w14091460_

Round 1
Reviewer 1 Report
This is an interesting paper, that shares experience of innovation management of a major DFPP. However, the paper is not suitable for publication in Water Journal due to two main issues. First, the paper is classified in the category of ‘Article’. As defined in the Water Journal website, “Articles” should contain original research manuscripts (“The journal considers all original research manuscripts provided that the work reports scientifically sound experiments and provides a substantial amount of new information.”) The paper does not contain sufficient information that can be considered and derived from a research. The content seems to be much more collective views of authors involved in the paper. Thus, this does not really demonstrate methodological rigour, raising issues with the validity and reliability of the findings/ claims made in the paper. Second, the contribution of the paper, particularly on existing theories (for example, in innovation management) is unclear. The paper has not sufficiently articulated the position of the paper against the extant of existing knowledge.
Further detailed observations:
- Introduction is unclear, and should contain the argument that provides potential contribution to knowledge. It contains unclear sentences and phrases. For example, ‘expresses the probability of flooding, towards adaptive delta management’ and ‘quick literature review’. What does ‘quick literature review’ mean? It does not reflect that sufficient rigour and comprehensive coverage of existing knowledge have been made. So, how can the readers be convinced with the quality of the literature review?
- Underpinning theory of Figure 1 should be strengthened; need to clarify these steps, how one step leads to the next? Why Maslow? Is it the most appropriate concept for this purpose?
- The goal on increasing production rate is seen as ‘effectiveness’, but it looks more ‘efficiency’.
- A number of grammatical and typo errors, and inaccurate use of terms, which compromise clarity and understanding of readers.
- Section 5 is interesting, but needs further development, particularly sub-section “Knowledge and innovation projects: developments”
- Figure 5 caption should be improved. Should the text be included in the main text, rather than in the Figure caption.
7. In the discussion section, there are few new findings/information which have not been included in the previous sections, including “DFPP organisation held a survey at the end of 2021 amongst everyone involved in the DFPP about her knowledge and innovation strategy”, “The first results of this survey showed that the respondents were innovation-minded and becoming more aware of the need to share knowledge.” Should this be part of the findings, and then discussed further in the discussion section?
Reviewer 2 Report
Good review summary of Netherlands flood control system. Here are my comments:
1. The Dutch regional water authorities must strengthen about 1,500 km of flood defences to meet the flood protection standards that have been set in 2017 (reference?). Those standards originate from adaptation of the Flood Risk Management Policy in 2015 (reference?).
2. The paper uses/build upon Flood Protection standards 2017 but the authors never explain what these standards are?
3. How about lessons learned from other countries? For instance, State of Florida in the United States has very similar condition to NL. In fact, there are lots of collaborations between these two.
4. Fig. 2 is blurry.
5. I was expecting to see more details of the plan for sea-level rise but the authors just note the existence of issue (#76) and not the plan.
6. I did not see any info on supervisory control and data acquisition (SCADA). The SCADA is a backbone of every automated canal network and provides real-time monitoring and control capabilities. A reliable network-wide SCADA system is critical in providing operators with the data and level of control necessary to meet their strategic cost goal, while maintaining their core mission and regulatory compliance.
Reviewer 3 Report
Main concerns:
The article looks like a report made for the Dutch authorities. It is not clear what its objective is, the methodology is confusing, and the results achieved are not clear. It is full of acronyms that make it difficult to read and understand. As currently written, it cannot be accepted for publication. If the authors can write it again, in such a way that it clearly shows what the objectives are, the methodology followed and the concrete results, the problems encountered and the lessons learned, the article can be accepted. Although it must be said that it is an unusual article. It is important to remember that the article must provide interest for the reader, either because of the novelty of the methodology or the results, or because it can reproduce the process in other conditions.
References does not show the same format. On some occasions the year is in bold letters, other ones between brackets, and other ones without brackets.
The greatest part of the references is not included in the paper. Please, eliminate them.
Minor changes
You introduce in the text a great number of institutions, commissions, flood plans and so on. Then, it is difficult to follow your argument. It will be useful that you add a flux diagram showing all these entities and plans and their links.
Line 17. Replace “and how she is” by “and how it is”
Line 57. Replace “20212-2018” by “2012-2018”
Line 58. Please clarify in the text what do you mean with “representatives of the triple helix”.
Line 59. Please, put a reference about the “Pyramid of Flood Risk Management framework” or introduce this concept. As you show it in Figure 1 you can refer in the text this figure. Could you introduce the Muslow reference at the foot of figure 1?
Line 69. Replace “The Dutch” by “the Dutch”
Line 118. Add a dot at the end of the sentence.
Line 129. Move the explanation of the acronym RWA from line 137 to here.
Line 205. Please, explain what TRL is.
Line 233. Add a parenthesis after “triggers”
Line 465. You introduce for first time the FODIKI methodology in the Discussion, and you say that it has been used. It would be interesting to introduce it in the main text, not only in the Discussion
Round 2
Reviewer 1 Report
The paper has undergone significant improvements from the first version. The paper, describing co-creation and diffusion of knowledge alongside/ with stakeholders in practical way, is worthy of publication. The authors should be applauded for their effort to revise the paper. Reading the paper for the second time, there are few suggestions for further improvements.
- It is essential to state the contribution to knowledge explicit, particularly in the discussion and conclusion section. This is an opportunity to re-connect with the existing literature/ knowledge. It is critical to state how the work has filled the gap, or contributed to knowledge explicitly.
- Significant parts of the paper need further rationalisation of the way which it can be best presented to the readers. Some are too long, and do not sufficiently focussed. For example, those paragraphs in lines 203 to 263, and section 3.2. Better structures should be considered for these sections. Not all need to be presented, but only selected ones that help better understanding of the main message (i.e. supporting your claim of contribution to knowledge). The authors are encouraged to spend more time on this aspect.
- The paper contains grammatical errors and repetitions. For example, issue on stability and piping, in line 206, which seems to be repeated in line 232. Lines 355-356, is ‘Monitoring took the form of a survey’ repeated by ‘Monitoring takes place by means of a survey’, just in the next sentence? These should be identified by second-read and proof-read the paper, properly.
- Need to explain monitoring survey more explicitly in the method section.
- Sentence in lines 434-436 should be re-assessed. Do the readers need to know? If yes, what is the added value of including this sentence?
Reviewer 3 Report
Thank you very much for the new version of your paper. It is a pleasure for me telling you that from my point of view your paper can be accepted to be published as it is.
Author Response
Thank you very much for this positive news. We would like to thank you for your critical and targeted comments to the earlier version of the article.